# Five-Year Trends in Low-Density Lipoprotein Cholesterol Management in a Primary Healthcare Centre in Kaunas

**DOI:** 10.3390/medicina60121963

**Published:** 2024-11-28

**Authors:** Gediminas Urbonas, Lolita Šileikienė, Leonas Valius, Evelina Grigalė, Vilius Kaupas, Tautvydas Juška, Gabrielė Vėbraitė, Ingrida Grabauskytė

**Affiliations:** 1Department of Family Medicine, Medical Academy, Lithuanian University of Health Sciences, 50161 Kaunas, Lithuania; leonas.valius@lsmu.lt (L.V.); evelina.ruzaite@stud.lsmu.lt (E.G.); vilius.kaupas@stud.lsmu.lt (V.K.); tautvydas.juska@stud.lsmu.lt (T.J.); 2Institute of Cardiology, Medical Academy, Lithuanian University of Health Sciences, 50103 Kaunas, Lithuania; lolita.sileikiene@lsmu.lt; 3Clinical Department of Internal Medicine, Lithuanian University of Health Sciences, 50103 Kaunas, Lithuania; gabriele.vebraite@stud.lsmu.lt; 4Department of Physics, Mathematics and Biophysics, Lithuanian University of Health Sciences, 50103 Kaunas, Lithuania; ingrida.grabauskyte@lsmu.lt

**Keywords:** diabetes, dyslipidemia, low-density lipoprotein cholesterol, cardiovascular diseases, primary healthcare

## Abstract

*Background and Objectives*: Low-density lipoprotein cholesterol (LDL-C) is a marker of cardiovascular risk and its management. This study evaluated LDL-C control trends in patients treated at a primary healthcare center in Lithuania. *Materials and Methods*: Five-year (2019–2023) data on patients aged 40 years or older diagnosed with dyslipidemia were extracted from a real-world data and analytics platform, TriNetX. Patients were grouped into three groups: patients with dyslipidemia only (control group), patients with dyslipidemia and diabetes, and patients with dyslipidemia and cardiovascular disease (CVD). The following LDL-C goals were used for analysis: <1.4 mmol/L (a goal for very-high-risk patients in primary or secondary prevention), <1.8 mmol/L (a goal for high-risk patients), and <3.0 mmol/L (a goal for low-risk patients). *Results*: There were 18,646 patients with dyslipidemia. Of them, 8.9% of patients had diabetes, and 3.1% of patients had CVD. The median LDL-C concentration was significantly lower in patients with diabetes (2.82 mmol/L, *p* < 0.05) and in patients with CVD (2.45 mmol/L, *p* < 0.05) than in the control group (3.35 mmol/L). A trend of decreasing median LDL-C over the years was observed in all groups, with the lowest median values in 2023. The proportion of patients with LDL-C levels < 3 mmol/L increased from 32.0% in 2019 to 41.5% in 2023. The proportion of diabetic patients achieving LDL-C < 1.8 mmol/L increased from 7.4% to 25.9%, and those achieving LDL-C < 1.4 mmol/L increased from 3.1% to 10.6%. The proportion of patients with CVD achieving LDL-C < 1.8 mmol/L increased from 14.2% to 36.6%, and those achieving LDL-C < 1.4 mmol/L increased from 3.0% to 14.0%. *Conclusions*: Trends in the control of LDL-C levels are positive over 5 years, but a significant proportion of patients still did not reach the recommended target levels.

## 1. Introduction

Despite the mortality due to cardiovascular diseases (CVDs) having decreased in recent decades, CVDs remain the main cause of mortality worldwide, accounting for approximately a third of all deaths [1]. In 2021, there were 1.71 million deaths from CVD in the European Union (EU), equivalent to 32.4% of all deaths [2]. Lithuania ranked 4th in the EU, with a CVD mortality of 48.7% [2].

Dyslipidemia, hypertension, smoking, overweight, and diabetes are the modifiable CVD risk factors, accounting together for more than 50% of CVD cases and 20% of deaths from any cause [3].

Low-density lipoprotein (LDL), together with other apolipoprotein B-containing lipoproteins, are causal to the development of atherosclerotic CVDs [4]. LDL cholesterol (LDL-C) serves as a marker of cardiovascular (CV) risk and its management [5]. Numerous clinical trials of lipid-lowering therapy (LLT) demonstrated a decrease in CVD incidence and mortality [6]. A meta-analysis of 38 randomized controlled trials of guideline-recommended LLTs (statins, ezetimibe, and proprotein convertase subtilisin-kexin type 9 inhibitors) reported that LDL-C reduction by 1 mmol/L resulted in a 22% lower risk for major vascular events (i.e., CV death, myocardial infarction, unstable angina, coronary revascularization, or stroke) [7]. Another meta-analysis (49 randomized controlled trials) reported similar risk reduction effects for statins and nonstatin interventions: the risk ratio for major vascular events per 1 mmol/L reduction in LDL-C level was 0.77 for statins and 0.75 for nonstatin interventions (i.e., diet, bile acid sequestrants, ileal bypass, and ezetimibe) [8].

Acknowledging the growing evidence of LDL-C’s critical role in the development of CVDs, clinical guidelines have progressively introduced more stringent lipid management goals over the last decades [9]. Current European clinical guidelines recommend reducing LDL-C to the target levels based on individual CV risk [10,11]. The 2019 European Society of Cardiology/European Atherosclerosis Society (ESC/EAS) guidelines for the management of dyslipidemias [10] and the more recent ESC guideline on cardiovascular disease prevention [11] recommend LDL-C reduction of at least 50% from baseline and an LDL-C goal of <1.4 mmol/L in very high-risk patients, LDL-C reduction of at least 50% from baseline and an LDL-C goal of <1.8 mmol/L in high-risk patients, and an LDL-C goal of <3.0 mmol/L in low-risk patients. As per ESC/EAS guidelines, individuals with diabetes (type 1 or type 2), atherosclerotic CVDs, chronic kidney disease, or very high levels of risk factors are considered to be at very high or high CV risk and require no further risk estimation [10].

Since primary healthcare plays an essential role in the identification of patients at risk, management of risk factors, and prevention of CVDs [12,13,14], we evaluated LDL-C control over a 5-year period in patients with dyslipidemia treated at a primary healthcare center in Lithuania.

## 2. Materials and Methods

This study was conducted at the primary healthcare center Saulės šeimos medicinos centras (Kaunas, Lithuania), which serves a total of 27,000 people.

Patients aged 40 years or older who were diagnosed with dyslipidemia (code E78 according to the International Classification of Diseases, 10th revision [ICD-10]) were eligible for this study. There were no exclusion criteria other than age below 40 years.

Within the study population, patients were grouped into three groups: (i) a control group consisting of patients with dyslipidemia only (no diabetes or CVD), (ii) patients with dyslipidemia and diabetes (ICD-10 codes E10 or E11), and (iii) patients with dyslipidemia and CVDs (ICD-10 codes I25.2, I21, I70, I63, or Z95). The ICD-10 codes were chosen following the definitions of CV risk categories recommended by the 2019 ESC/EAS guidelines [10]. Patients with unstable angina, stable angina, transient ischemic attack, or peripheral arterial disease were not included in the CVD group because of insufficient data to verify these diagnoses. Patients having both diabetes and CVD diagnoses were included in both groups.

The LDL-C goals set by the 2019 ESC/EAS guidelines [10] and the 2021 ESC guidelines [11] were used for analysis: <1.4 mmol/L, <1.8 mmol/L, and <3.0 mmol/L.

Data on patients’ diagnosis, sex, age, and LDL-C concentration in the period from 2019 to 2023 were extracted from TriNetX, a real-world data and analytics platform. As a part of routine clinical care of patients with dyslipidemia, LDL-C concentrations were measured at a certified analytical laboratory contracted by the health care center. For patients having more than one LDL-C measurement within a calendar year, the latest value was used for analysis.

Data were analyzed using the statistical software package SPSS version 29. The normality of distribution was assessed with the Kolmogorov–Smirnov test. Quantitative variables were presented as mean (standard deviation) in case of normal distribution or as median (first quartile and third quartile or interquartile range) in case of non-normal distribution. Qualitative data were presented as a number (n) and percentage (%). A nonparametric Kruskal–Wallis test was used to cosmpare quantitative variables among more than two groups. A Pearson’s chi-square test was used to compare qualitative variables. The annual percent change (APC), together with the corresponding 95% confidence interval, was calculated to describe temporal trends in the proportion of patients achieving LDL-C goals across the study groups.

Univariate and multivariate logistic regression analyses were performed to identify independent variables that were associated with LDL-C values of <3.0 mmol/L. The dependent variable was LDL-C (<3.0 mmol/L versus ≥3.0 mmol/L). The independent variables used in the univariate analysis were age, sex, diagnosis of diabetes, and CVDs. Multivariate logistic regression was employed using the enter method, i.e., all variables that were significant in univariate analysis were entered into the model at the same time.

Statistically significant (two-sided *p* < α) differences were interpreted at a 5% level of significance (i.e., α = 0.05).

## 3. Results

The study population consisted of 18,646 patients with dyslipidemia (one patient was excluded because of missing information on sex). There were 8.9% of patients with type 1 or type 2 diabetes mellitus and 3.1% of patients with CVDs. Detailed information on the number of patients with specific ICD-10 codes is provided in Table 1.

The proportion of women was slightly higher (58.2%). The mean age was 58.1 years (range: 40 to 89 years).

In the whole study population, the median LDL-C concentration was 3.29 mmol/L, and 38.5% of patients had LDL-C < 3 mmol/L (Table 1).

Within the whole 5-year period, the median (interquartile range) LDL-C concentration was significantly lower in patients with diabetes (2.82 [1.56] mmol/L, *p* < 0.001) and in patients with CVD (2.45 [1.53] mmol/L, *p* < 0.001) than in the control group (3.35 [1.34] mmol/L).

A general tendency of decreasing median LDL-C over the years was observed in all groups, with the lowest median values in each study group in 2023 (Table 2).

When analyzing data for the whole 5-year period, the proportion of patients achieving the target LDL-C levels was significantly higher among patients with diabetes or CVDs compared with the patients with dyslipidemia only (control group) (Table 3).

The analysis of LDL-C target achievement in each year showed that the proportion of patients achieving LDL-C of <3.0 mmol/L in the patients with dyslipidemia only (control group) was similar in 2019–2022, but it was significantly higher in 2023 than in the previous years (Table 4). Among patients with diabetes and CVDs, the proportion of patients achieving LDL-C of <1.8 mmol/L or <1.4 mmol/L increased (with some fluctuations) over the years, but the most notable increase was also observed in 2023 (Table 4).

As shown by the APC, the proportion of patients achieving LDL-C goals increased at an average rate ranging from 15.7% to 47.5% annually across the different study groups (Table 4).

In the univariate analysis, age, male sex, diagnosis of diabetes, or CVD were all associated with higher odds ratios to reach LDL-C values of <3.0 mmol/L (Table 5). All these variables remained significant in the multivariate analysis (Table 5).

## 4. Discussion

The control of LDL-C levels in a single primary healthcare center improved over a five-year period. In the whole group of patients with dyslipidemia, the proportion of patients with LDL-C levels < 3 mmol/L increased from 32.0% in 2019 to 41.5% in 2023. The proportion of patients achieving more stringent goals set for high- and very high-risk groups also increased. The achievement of LDL-C < 1.8 mmol/L increased 2.6-fold in patients with dyslipidemia and CVDs and 3.5-fold in patients with dyslipidemia and diabetes. Similarly, the achievement of LDL-C < 1.4 mmol/L increased 4.7-fold in patients with dyslipidemia and CVDs and 3.4-fold in patients with dyslipidemia and diabetes.

Recent data on LDL-C goal achievement in Lithuania are scarce. In the Lithuanian arm of an EUROASPIRE V (European Action on Secondary and Primary Prevention by Intervention to Reduce Events) survey, only 4.5% of primary prevention patients reached their risk-based 2019 ESC/EAS LDL-C goal (21% of the study population used statins) [15]. Earlier data from the Lithuanian High Cardiovascular Risk primary prevention program revealed that in 2009–2015, 13.5% of middle-aged individuals without overt CVD had severe dyslipidemia, defined as total cholesterol ≥ 7.5 mmol/L, or LDL-C ≥ 6 mmol/L, or triglycerides ≥ 4.5 mmol/L [16]. A study on risk factor trends and mortality from noncommunicable diseases in Lithuania observed a significant reduction in mean levels of LDL-C and total cholesterol between 1986 and 2008 [17]. In all three Baltic countries, less than 20% of primary care patients receiving LLT had LDL-C at the target level defined according to the 2011 ESC/EAS guideline [18].

Despite the positive trends observed in our study, the overall control of LDL-C levels remains far from optimal, with a significant proportion of patients still not reaching the recommended target levels. Real-world data from other European countries also shows insufficient dyslipidemia control, even in patients receiving LLT [19]. An EU-wide DA VINCI study found that only 33% of patients receiving LLT achieved their risk-based 2019 ESC/EAS LDL-C goal, and goal achievement was even lower among patients at higher risk [20]. As per the EUROASPIRE V survey, 46.9% of treated dyslipidemia patients attained an LDL-C target of <2.6 mmol/L [21]. In high/very high-CV-risk patients in Germany (36.3% of them were treated with LLT), 7.2% of patients attained an LDL-C of <1.8 mmol/L, and 22.8% of patients had an LDL-C from 1.8 to <2.6 mmol/L [22]. In Portugal, a population-based cohort study analyzed electronic health records of patients followed in both primary and secondary care and reported that LDL-C targets as per the 2019 ESC/EAS guidelines were achieved in only 44%, 27%, 7%, and 3% of low, intermediate, high, and very high-CV-risk patients, respectively. Of note, the study population included both LLT-treated and untreated patients [23]. A SANTORINI study of patients with high and very high CV risk across 14 Central and Western European countries also reported a low proportion of patients (20.1%) achieving risk-based LDL-C goals, as per the 2019 ESC/EAS guidelines. More than half of these patients were receiving LLT monotherapy, and about one-fourth of patients were on combination LLT [24]. Even more concerning results were reported in a recent Belgium study, in which only 18% of patients with established atherosclerotic CVDs receiving LLT had an LDL-C <1.4 mmol/L [25].

In this study, we did not analyze the trends in LLT use among patients with dyslipidemia. A significant increase in statin prescriptions (from 8.28 defined daily doses for a thousand inhabitants per day) in 2010 to 96.06 in 2021 was reported in Lithuania [26]. A similar increase in the use of LLT could be expected in our study population over the 5-year period. Increasing awareness of cardiovascular risk management among healthcare providers and patients could also play a role in favorable trends observed in our study.

The finding that the most notable achievement of LDL-C goals was observed in 2023 is noteworthy. At the beginning of 2023, a few significant changes in the national cardiovascular disease prevention program were implemented in Lithuania. These changes included extending age limits for a target population, risk-based frequency of medical examination, and financing of the program [27]. In addition, the reimbursement criteria for ezetimibe, a selective inhibitor of the intestinal absorption of cholesterol, were changed, allowing for its prescription to a larger population of patients (i.e., those with LDL-C ≥ 1.4 mmol/L despite the use of maximal tolerable doses of statins for at least 4 weeks). However, it is too early to speculate if these changes might have had such an immediate effect on LDL-C control levels, and further observations of dyslipidemia management trends are necessary to assess the impact of these governmental decisions.

Of note, the study period included the years of the COVID-19 pandemic. Most likely, this caused the drop in study patients (i.e., patients with dyslipidemia having at least one LDL-C measurement) in 2020. After that, the number of patients steadily increased, starting in 2021. Nevertheless, the positive trend of decreasing median LDL-C concentrations and the corresponding increase in the proportion of patients who achieved LDL-C goals seemed to not be affected by the pandemic.

Real-world evidence implies that LDL-C goals recommended by the current clinical guidelines are difficult to attain with LLT monotherapy [23]; therefore, less stringent reimbursement conditions facilitating access to LLT other than statins and thus the possibility to prescribe combination therapy are necessary to improve LDL-C control.

Our study has several limitations. We analyzed single-center data; therefore, our findings cannot be generalized to other healthcare centers or to the whole country. In addition, we defined study groups based on rather rough estimates of CV risk (i.e., presence or absence of type 1 or type 2 diabetes or CVD diagnosis). Thus, our control group might have at least some patients at high or very high risk (e.g., patients with unstable angina, stable angina, transient ischemic attack, peripheral arterial disease, chronic kidney disease, or other significant CVD risk factors). However, it is unlikely that the proportion of potentially misclassified patients differed by year; therefore, the observed positive trends in LDL-C control are not biased.

Despite these limitations, this analysis of medical records of the whole population of patients with dyslipidemia treated at a single primary healthcare center provides valuable insights into LDL-C management in a real-world setting. This helps to identify the gaps between the recommended LDL-C target levels and the actual rates of achieving these targets in routine care, as well as trends over time.

The positive trends in LDL-C control observed over the 5-year period in our health care center are encouraging. While our study covered a relatively short period, the consistent increase in the proportion of patients achieving LDL-C goals suggests a potential for reductions in major adverse cardiovascular events in the long term. Further research with extended follow-up is needed to confirm whether these improvements in LDL-C control translate into measurable declines in cardiovascular event rates over time.

## 5. Conclusions

Analysis of LDL-C control in patients treated at a single healthcare center in Lithuania demonstrated positive trends in the control of dyslipidemia. However, a significant proportion of patients still did not reach the target levels recommended by current clinical guidelines, which highlights the necessity for enhanced efforts to further improve LDL-C management.

## Figures and Tables

**Table 1 medicina-60-01963-t001:** Demographic and clinical characteristics of the study population.

Characteristics	
Women, *n* (%)	10,848 (58.2)
Men, *n* (%)	7798 (41.8)
Age, mean (standard deviation), years	58.1 (12.3)
LDL-C concentration	
median (Q1, Q3), mmol/L	3.29 (2.6, 4.0)
<3 mmol/L, *n* (%)	7188 (38.5)
<1.8 mmol/L, *n* (%)	1219 (6.5)
<1.4 mmol/L, *n* (%)	388 (2.1)
Patients with dyslipidemia (no diabetes or CVDs), *n* (%)	16,268 (87.2)
Patients with diabetes mellitus and CVDs, *n* (%)	137 (0.7)
Patients with diabetes mellitus, *n* (%)	1662 (8.9)
Patients with CVDs, *n* (%)	579 (3.1)

CVD, cardiovascular disease; LDL-C, low-density lipoprotein cholesterol; Q, quartile.

**Table 2 medicina-60-01963-t002:** Median LDL-C concentrations: comparison among years in each study group.

Study Group	2019	2020	2021	2022	2023	*p* Value ^a^
Dyslipidemia only	*n* = 2750	*n* = 2509	*n* = 3448	*n* = 3748	*n* = 3813	
	3.42 (1.31)	3.39 (1.31)	3.40 (1.32) ^b^	3.41 (1.33)	3.21 (1.36) ^c^	<0.001
Dyslipidemia + diabetes	*n* = 257	*n* = 313	*n* = 348	*n* = 396	*n* = 348	
	3.20 (1.69)	2.93 (1.46) ^b^	2.79 (1.48) ^b^	2.87 (1.65) ^b^	2.54 (1.42) ^c^	<0.001
Dyslipidemia + CVD	*n* = 134	*n* = 96	*n* = 116	*n* = 140	*n* = 93	
	2.62 (1.49)	2.60 (1.65)	2.35 (1.58)	2.39 (1.48)	2.17 (1.36) ^b^	0.008

Values are medians (interquartile range). CVD, cardiovascular disease. ^a^
*p*-value for difference among years in a given study group. *p* values for differences between specific years in a given study group: ^b^
*p* < 0.05 vs. 2019. ^c^
*p* < 0.05 vs. 2019, 2020, 2021, 2022.

**Table 3 medicina-60-01963-t003:** The distribution of patients achieving different LDL-C targets in study groups in 2019–2023.

LDL-C Target	Dyslipidemia Only	Dyslipidemia + Diabetes	Dyslipidemia + CVDs	*p* Value ^a^
	*n* = 16,268	*n* = 1662	*n* = 579	
<3.0 mmol/L, *n* (%)	5762 (35.4)	930 (56.0) ^b^	401 (69.3) ^b^	<0.001
<1.8 mmol/L, *n* (%)	755 (4.6)	274 (16.5) ^b^	138 (23.8) ^b^	<0.001
<1.4 mmol/L, *n* (%)	211 (1.3)	110 (6.6) ^b^	50 (8.6) ^b^	<0.001

CVD, cardiovascular disease; LDL-C, low-density lipoprotein cholesterol. ^a^
*p*-value for difference among study groups. ^b^
*p* < 0.05 vs. control group (patients with dyslipidemia only).

**Table 4 medicina-60-01963-t004:** Distribution of patients achieving different LDL-C targets and annual percent change: comparison among years in each study group.

Study Group	2019	2020	2021	2022	2023	*p* Value ^a^	APC, % (95% CI)
LDL-C < 3.0 mmol/L							
Dyslipidemia only, *n* (%)	*n* = 2750	*n* = 2509	*n* = 3448	*n* = 3748	*n* = 3813		
	881 (32.0)	848 (33.8)	1209 (35.1)	1243 (33.2)	1581 (41.5) ^c^	<0.001	15.7 (9.7–21.8)
Dyslipidemia + diabetes, *n* (%)	*n* = 257	*n* = 313	*n* = 348	*n* = 396	*n* = 348		
	115 (44.7)	166 (53.0)	201 (57.8) ^b^	209 (52.8)	239 (68.7) ^c^	<0.001	20.1 (13.4–26.7)
Dyslipidemia + CVDs, *n* (%)	*n* = 134	*n* = 96	*n* = 116	*n* = 140	*n* = 93		
	91 (67.9)	59 (61.5)	79 (68.1)	101 (72.1)	71 (76.3)	0.225	−6.02 (−21.1–9.1)
LDL-C < 1.8 mmol/L							
Dyslipidemia only, *n* (%)	*n* = 2750	*n* = 2509	*n* = 3448	*n* = 3748	*n* = 3813		
	93 (3.4)	110 (4.4)	163 (3.9)	147 (3.9)	242 (6.3) ^c^	<0.001	27.0 (18.0–36.0)
Dyslipidemia + diabetes, *n* (%)	*n* = 257	*n* = 313	*n* = 348	*n* = 396	*n* = 348		
	19 (7.4)	44 (14.1)	50 (14.4)	71 (17.9) ^b^	90 (25.9) ^d^	<0.001	47.5 (34.8–60.3)
Dyslipidemia + CVDs, n (%)	*n* = 134	*n* = 96	*n* = 116	*n* = 140	*n* = 93		
	19 (14.2)	18 (18.8)	28 (24.1)	39 (27.9)	34 (36.6) ^b^	0.001	15.7 (4.3–27.0)
LDL-C < 1.4 mmol/L							
Dyslipidemia only, *n* (%)	*n* = 2750	*n* = 2509	*n* = 3448	*n* = 3748	*n* = 3813		
	15 (0.5)	30 (1.2)	43 (1.2) ^b^	53 (1.4) ^b^	70 (1.8) ^b^	<0.001	47.0 (36.6–57.4)
Dyslipidemia + diabetes, *n* (%)	*n* = 257	*n* = 313	*n* = 348	*n* = 396	*n* = 348		
	8 (3.1)	19 (6.1)	15 (4.3)	31 (7.8)	37 (10.6) ^e^	<0.001	46.7 (29.9–63.7)
Dyslipidemia + CVDs, *n* (%)	*n* = 134	*n* = 96	*n* = 116	*n* = 140	*n* = 93		
	4 (3.0)	7 (7.3)	10 (8.6)	16 (11.4)	13 (14.0) ^b^	0.034	34.3 (19.0–49.5)

APC, annual percent change; CI, confidence interval; CVD, cardiovascular disease; LDL-C, low-density lipoprotein cholesterol. ^a^
*p*-value for difference among years in a given study group. *p*-values for differences between specific years in a given study group: ^b^
*p* < 0.05 vs. 2019, ^c^
*p* < 0.05 vs. 2019, 2020, 2021, 2022, ^d^
*p* < 0.05 vs. 2019, 2020, 2021, ^e^
*p* < 0.05 vs. 2019, 2021.

**Table 5 medicina-60-01963-t005:** Univariate and multivariate logistic regression analysis for predicted LDL-C values of <3.0 mmol/L.

Variable	Univariate Analysis			Multivariate Analysis		
	Odds Ratio	95% CI	*p* Value	Odds Ratio	95% CI	*p* Value
Age	1.009	1.007–1.012	<0.001	1.003	1.000–1.005	0.046
Sex (male)	1.120	1.055–1.189	<0.001	1.102	1.035–1.174	0.002
Diabetes	2.296	2.080–2.533	<0.001	2.165	1.956–2.398	<0.001
Cardiovascular disease	3.786	3.221–4.450	<0.001	3.423	2.898–4.043	<0.001

CI, confidence interval.

## Data Availability

The data presented in this study are available upon request from the corresponding author due to privacy and legal reasons.

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
