# Peer review of "Five-Year Trends in Low-Density Lipoprotein Cholesterol Management in a Primary Healthcare Centre in Kaunas"

_medicina, 2024, doi:10.3390/medicina60121963_

Round 1
Reviewer 1 Report
Comments and Suggestions for Authors
Thank you for the opportunity to review of manuscript entitled "Trends in the Management of Low-Density Lipoprotein Cholesterol in the Primary Healthcare Centre in Kaunas over the Five-Year Period".
I read the manuscript with great attention because I believe that the topic is worthy of it. I congratulate the authors for undertaking this task.
The abstract is readable and meets the criteria for proper composition - I have no issues aside from its excessive length (about 324 words), which will likely necessitate condensation.
The introduction briefly introduces the reader to the further parts of the manuscript. I have no objections to this section.
In the methodological section:
1) The authors pointed out that the criteria for excluding participation in the study included: age below 40 years, oncological disease, severe autoimmune disease, palliative care patients, and pregnancy. This is a very rough approach to the topic - I recommend that the authors elaborate on this information (specifying which severe autoimmune diseases were identified in the study cohort) and incorporate a diagram illustrating the variability in the number of patients analyzed based on qualification and exclusion criteria (lines 86-87);
2) Did none of the patients in the analysis meet the criteria for both the second group (with diabetes) and the third group (with CVD)? If so, to which group were these patients classified? (lines 89-90);
3) Didn't the second group (with diabetes) also include patients with ICD-10 code E12, E13 or E14? If this is the case, could you please explain your reasoning, particularly in relation to E13 and E14? (line 90);
4) Why were patients with a history of pulmonary embolism (I26), cardiomyopathies (I42-I43), arrhythmias (I44-49), heart failure (I50) and others not included in the third group, since the diagnosis of atherosclerosis was taken into account? The authors should decide this situation either to include only acute conditions or to include all cardiovascular diseases that may be affected by dyslipidemia (line 91)
I have no objections to the Results section; however, the authors' unclear position regarding patients meeting both the criteria for diabetes and CVD should be kept in mind, as well as under-recognition (or over-recognition, depending on the approach) of certain CVDs.
I have no objections to the Discussion and Conclusions sections, taking into account previous comments (which may be relevant to the content of the mentioned sections). It may be necessary to elaborate on the paragraph addressing the study's limitations.
I believe that the work has the potential to be a good approach to the topic of proper management for patients with lipid-lowering treatment, but the manuscript necessitates refinement.
Despite the above, I congratulate the authors on taking on this topic and wish them further success in their scientific careers.
Reviewer 2 Report
Comments and Suggestions for Authors
The following are my concerns.
1. It is not clear what the authors mean by “LDL-C goals”. These need to be better described and explained why they were chosen and how they were achieved or evaluated.
2. Whether these are recommended reference/normal ranges of LDL-C for each of the groups, or they are the levels the patients are trying to achieve or target in order to be able to lead a better life given their health conditions, and if yes, how are they doing this? Nutrition, exercise, lifestyle, medication intervention? It is not described or clear.
3. Were the same group of people followed over the five year period?
4. Was the higher number of the dyslypedimia patients due to more reporting after the pandemic? If not, what could be the reason for this increase in numbers?
5. While some interventions like. LLT and prescription drugs are mentioned in the discussion, it would be helpful to explain if these were interventions being evaluated.
6. How are the patients managing their LDL-C levels? Were there interventions to allow the changes in LDL-C or was the study trying to study natural progression of LDL-C levels in these individuals and the outcomes over time?
7. The study covers the period of the pandemic. With that in mind, there would certainly be an impact on the patients being able to get evaluated. There is no discussion or mention of this. It is interesting that levels of LDL-C dropped, or reduced during those time, but authors do not consider how the pandemic situation may have affected the patients and their ability to get tested or evaluated.
8. How was the LDL-C measured? There is not mention of it in the methods. If it was done in hospital or labs, was it being performed during the pandemic too?
9. Presentation of the results could be improved by having more visual graphic representation.
I feel the authors can improve the manuscript by communicating better the approach of the study, as well as the significance and relevance.
Reviewer 3 Report
Comments and Suggestions for Authors
Dear authors
The title is clear and indicating the study's focus on LDL-C management trends but suggests to consider shortening it while remaining key elements. For example, "Five-Year Trends in LDL-C Management in a Primary Healthcare Center in Kaunas".
In the conclusion in abstract suggests discuss implications for practice or future research to enhance the abstract's impact.
In the Methods describes the study design clearly and mention the analysis methods briefly.
In line 98” Normality of distribution was assessed with the Kolmogorov-Smirnov test for samples larger than 50 and the Shapiro-Wilk test for samples between 30 and 50”you have not sample size less than 50!
According to the objectives of the study, detailed information on the number of patients with specific ICD-10 codes in Table 1 is unnecessary
Title of table 1 is sociodemographic but you have not this variables!!!!
Use figures where appropriate to present trend more visually. This can enhance reader comprehension.
You can calculate annual percent changes(APC) for the trend
Table 2 is . Median LDL-C concentrations but in table text you writ “n (%)” What is in parentheses?
Will people be followed in 2019 to 2023? If yes, why has the number increased in Table 2 in 2021? Is a survey done on different people every year?
Lin 135: whole 2-year period, OR 5 years?
Footnoot of table 3 “. a p value for difference 139 among study groups. B p < 0.05 vs control group (patients with dyslipidemia only). It is not necessary at all because you have not b
It is not correct to use “Number (%) “ in the title of the table, use distribution instead
Why didn't you include groups (control group, patients with dyslipidaemia and diabetes, and patients with dyslipidaemia and CVD) as independent variables instead of diagnosis of diabetes or CVD?
Were these people all taking medication? What is your interpretation of the effect of taking medication on reducing LDL?
Line 107 “The dependent variable was LDL-C (< 3.0 mmol/L versus ≥ 3.0 mmol/L)”
Why did you have three target groups in the method, but you changed it to two groups? Why didn't you use ordinal logistics for 3 groups?
Line 96” patients having more than one LDL-C measurement within a calendar year, the latest value was used for analysis.” It would not be better to average for people with more measurements
Discuss the broader context of your findings. What do these trends mean for healthcare practice in Lithuania or similar settings?
Round 2
Reviewer 1 Report
Comments and Suggestions for Authors
Thank you for the opportunity to do the second review article entitled "Five-Year Trends in Low-Density Lipoprotein Cholesterol Management in a Primary Healthcare Centre in Kaunas".
The authors have responded to the comments from my last review; the majority of their responses are acceptable.
Author Response
Thank you for valuable comments.
Reviewer 2 Report
Comments and Suggestions for Authors
The comments and suggestions have been addressed by the authors.
Author Response
Thank you for valuable comments.
Reviewer 3 Report
Comments and Suggestions for Authors
1- Pleas introduce the reason of new group ldl (2group:response to comment 15 ) in methode section.
2- Mention the lack of drug analysis as a limitation in discussion section.
Author Response
Thank you for valuable comments. The article has been updated accordingly.